# Paired electrolysis-enabled nickel-catalyzed enantioselective reductive cross-coupling between α-chloroesters and aryl bromides

Dong Liu[1,3], Zhao-Ran Liu[1,3], Zhen-Hua Wang[1,3], Cong Ma[1], Simon Herbert[2], Hartmut Schirok[2] & Tian-Sheng Mei [1] ✉

Electrochemical asymmetric catalysis has emerged as a sustainable and promising approach to the production of chiral compounds and the utilization of both the anode and cathode as working electrodes would provide a unique approach for organic synthesis. However, precise matching of the rate and electric potential of anodic oxidation and cathodic reduction make such idealized electrolysis difficult to achieve. Herein, asymmetric cross-coupling between α-chloroesters and aryl bromides is probed as a model reaction, wherein alkyl radicals are generated from the α-chloroesters through a sequential oxidative electron transfer process at the anode, while the nickel catalyst is reduced to a lower oxidation state at the cathode. Radical clock studies, cyclic voltammetry analysis, and electron paramagnetic resonance experiments support the synergistic involvement of anodic and cathodic redox events. This electrolytic method provides an alternative avenue for asymmetric catalysis that could find significant utility in organic synthesis.

Asymmetric synthetic electrochemistry has a long history that was long limited by its reliance on ionic additives and a focus on the transfer of electrons from the electrode directly to the organic substrate[1–5]. Early effort toward asymmetric synthetic electrochemical methods was focused on leveraging chiral solvents, electrolytes, electrodes, mediators, or auxiliaries[1]. Later, chiral organocatalysts were introduced as sub-stoichiometric alternatives[6–11], and more recently, transition metal catalysis has been applied to electrochemical syntheses of valuable chiral compounds either by anodic oxidation or cathodic reduction[12–23]. By and large, in these processes, either hydrogen is generated, or a sacrificial anode is employed to ensure electroneutrality (Fig. 1a). Alternatively, paired electrolysis in undivided cells provides a more practical, atom-economical, and energy-efficient approach since both electrodes do work directly necessary for product formation[24–28]. However, paired electrolysis requires the rate and electric potential of the anodic oxidative reaction and the cathodic reductive reaction to be nearly perfectly matched[29–39]. Additionally, the heterogeneous electron transfer from the electrode to a substrate makes this process more

challenging[40,41]. To this end, the merger of asymmetric electrochemical catalysis with paired electrolysis is desired, however, it remains a significant challenge in electrochemical synthesis.

α-Arylated carbonyls are a class of useful scaffolds found widely in natural products and synthetic medicines[42–44]. Generally, these compounds are synthesized by asymmetric transition metal-catalyzed enolate arylation with the assistance of a strong base[45–51]. Alternatively, it could be obtained by transition metal-catalyzed enantioselective cross-coupling of α-chloroesters with organometallic reagents[52–56]. Recently, asymmetric reduction coupling of two electrophiles with chemical reductants to turn over the catalyst allowed the synthesis of α-arylated carbonyls under relatively mild reaction conditions[57,58]. Walsh and Mao reported a photo-induced nickel-catalyzed asymmetric reductive coupling of α-chloro esters with aryl iodides with Hantzsch ester as the terminal reductant[57]. With their own developed chiral BiOX ligand, Reisman and co-workers demonstrated nickel-catalyzed asymmetric reductive cross-coupling of α-chloroesters and (hetero) aryl iodides with Mn powder as reductant[58]. However, the reductive

[1]State Key Laboratory of Organometallic Chemistry, Shanghai Institute of Organic Chemistry, University of Chinese Academy of Sciences, CAS, Shanghai, China. [2]Pharmaceuticals, Research and Development, Bayer AG, 13353 Berlin, Germany. [3]These authors contributed equally: Dong Liu, Zhao-Ran Liu, Zhen-Hua Wang. ✉e-mail: mei7900@sioc.ac.cn

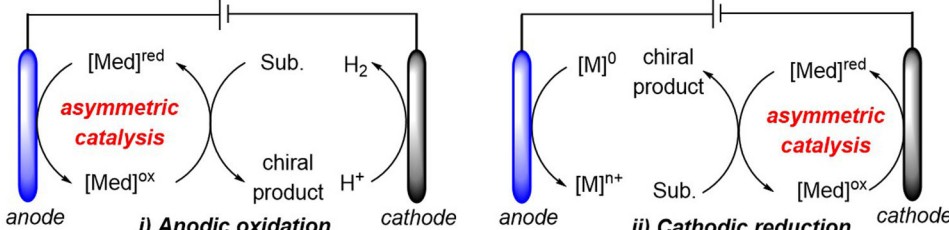

**Fig. 1 | Electrochemical asymmetric catalysis. a** Traditional strategies for electrochemical asymmetric catalysis. **b** Paired electrolysis for electrochemical asymmetric catalysis. **c** This work.

cross-coupling of aryl bromides with α-chloroesters remains a significant challenge, presumably owing to the lower reactivity of aryl bromide with the nickel catalyst compared with aryl iodides[57,58].

Pursuant to our interest in synthetic organic electrochemistry[59,60], we envisioned that the merger of paired electrolysis and asymmetric nickel catalysis could provide an alternative strategy for catalytic enantioselective reductive couplings of alkyl halides and aryl halides (Fig. 1b). Because of its low oxidation potential compared to the reactants[61], the halide could be preferentially oxidized at the anode to create a halogen radical[62,63]. Hydrogen atoms from a silane could be rapidly abstracted by the halogen radical to generate a silyl radical, which may subsequently abstract a halogen atom from an alkyl halide and generate an alkyl radical[64–67]. Simultaneously, cathodic reduction of [Ni$^{II}$] to [Ni$^{I}$] enables oxidative addition of aryl halide to give intermediate [ArNi$^{III}$X], which traps an alkyl radical to give [ArNi$^{III}$RX], from which reductive elimination affords an α-arylated cross-coupling product[68–70]. Herein, we reported the example of paired electrolysis-enabled enantioselective reductive cross-coupling between aryl bromides and α-chloroesters by using a chiral bis-imidazoline (BiIM) ligand (Fig. 1c).

## Results

### Optimization studies

Initially, the racemic reaction was optimized using methyl 4-bromobenzoate (**1a**), α-chloro ester (**1b**), NiBr$_2$•glyme (10 mol%), di-$^t$Bubpy (15 mol%), tris(trimethylsilyl)silane (TTMSS) (2.0 equiv), 2,6-lutidine (2.0 equiv), $n$-Bu$_4$NBF$_4$ (1.0 equiv) and DMAc (2.0 mL) under constant-current electrolysis at 6.0 mA for 6.0 h at room temperature, giving 90% isolated yield (Table 1, entry 1) (see Supplementary Table 1 for details). Next, we studied electrochemical

asymmetric reactions by investigating various chiral bioxazoline (BiOX) ligands according to previous reports on asymmetric reductive cross-couplings[57,58]. To our delight, Bn-BiOX (**L1**) could give 44% enantiomeric excess (ee) (entry 2). Further investigation of the BiOX family of ligands (entries 3–6) found that $s$-Bu-BiOX (**L4**) performed best, affording 77% ee with a 25% yield (entry 5). Then, we opted to modify the ester OR group to further improve the enantioselectivity. Unfortunately, replacing the ethyl group with PMP, O$^i$Pr, and O$^t$Bu groups gave similar results (entries 7–9). Inspired by the work of Nakamura[71], we increased the steric bulk of the ester OR substituent to 2,3,3-trimethylbut-2-yl, affording the desired product in 84% ee (entry 10). In addition, 4-heptyl substituted BiOX ligand (**L6**) gave 89% ee, although the yield was still low (entry 11)[72,73]. As the structural analog, chiral Bi-imidazoline (BiIM) ligands are also often employed in asymmetric Ni catalysis[74–84]. Therefore, we changed the sec-butyl and 4-heptyl substituted BiOX to corresponding chiral BiIM ligands (**L7** and **L8**), affording the desired product in 90% ee with 30% yield and 84% ee with 25% yield, respectively (entries 12 and 13). To our delight, under a cell potential of 2.9 V, with 3.0 equivalents of lutidine and 3.5 equivalents of TTMSS, the desired product **2a** could be isolated in 91% ee with 66% yield (entry 14). Control experiments indicated that the electricity, TTMSS, and catalyst are necessary to succeed in this transformation (entries 15 and 16). Replacement of α-chloro esters with α-bromoesters resulted in a low yield of the desired product, but good enantioselectivity (91%) was obtained (Supplementary Table 8).

### Substrate scope

With suitable reaction conditions in hand, the substrate scope was investigated to probe the generality and to identify the limitations of

**Table 1 | Reaction optimization and control studies[a]**

| Entry[a] | Ligand | R | Yield (%) | ee (%) |
|---|---|---|---|---|
| 1 | di-tBubpy | Et | 91 (90)[b] | - |
| 2 | L1 | Et | 10 | 44 |
| 3 | L2 | Et | 20 | 72 |
| 4 | L3 | Et | 37 | 46 |
| 5 | L4 | Et | 25 | 77 |
| 6 | L5 | Et | <5 | - |
| 7 | L4 | PMP | 20 | 67 |
| 8 | L4 | iPr | 15 | 76 |
| 9 | L4 | tBu | 15 | 77 |
| 10 | L4 | 2,3,3-Trimethylbut-2-yl | 10 | 84 |
| 11 | L6 | 2,3,3-Trimethylbut-2-yl | 11 | 89 |
| 12 | L7 | 2,3,3-Trimethylbut-2-yl | 30 | 90 |
| 13 | L8 | 2,3,3-Trimethylbut-2-yl | 25 | 84 |
| 14 | L7 | 2,3,3-Trimethylbut-2-yl | 66[c] | 91 |
| 15 | no electric current or TTMSS | 2,3,3-Trimethylbut-2-yl | <5 | - |
| 16 | No NiBr₂glyme or L7 | 2,3,3-Trimethylbut-2-yl | <5 | - |

[a]Reaction conditions: **1a** (0.2 mmol), (*rac*)-**1b** (2.0 equiv), NiBr₂·glyme (10.0 mol%), Ligand (15.0 mol%), 2,6-lutidine (2.0 equiv), TTMSS (2.0 equiv), *n*-Bu₄NBF₄ (1.0 equiv), and DMAc (2.0 mL) at rt, in an undivided cell subjected to 6.0 mA of current for 6.0 h using platinum anode (1.0 cm ×1.0 cm) and Ni foam cathode (2.0 × 3.0 cm²), argon. The yield was determined by ¹H NMR using CH₂Br₂ as an internal standard. Enantioselectivities were determined by chiral HPLC analysis.

[b]Isolated yield in parentheses.

[c]Lutidine (3.0 equiv), Tris(trimethylsilyl)silane (3.5 equiv), U$_{cell}$ = 2.9 V for 6.0 h.

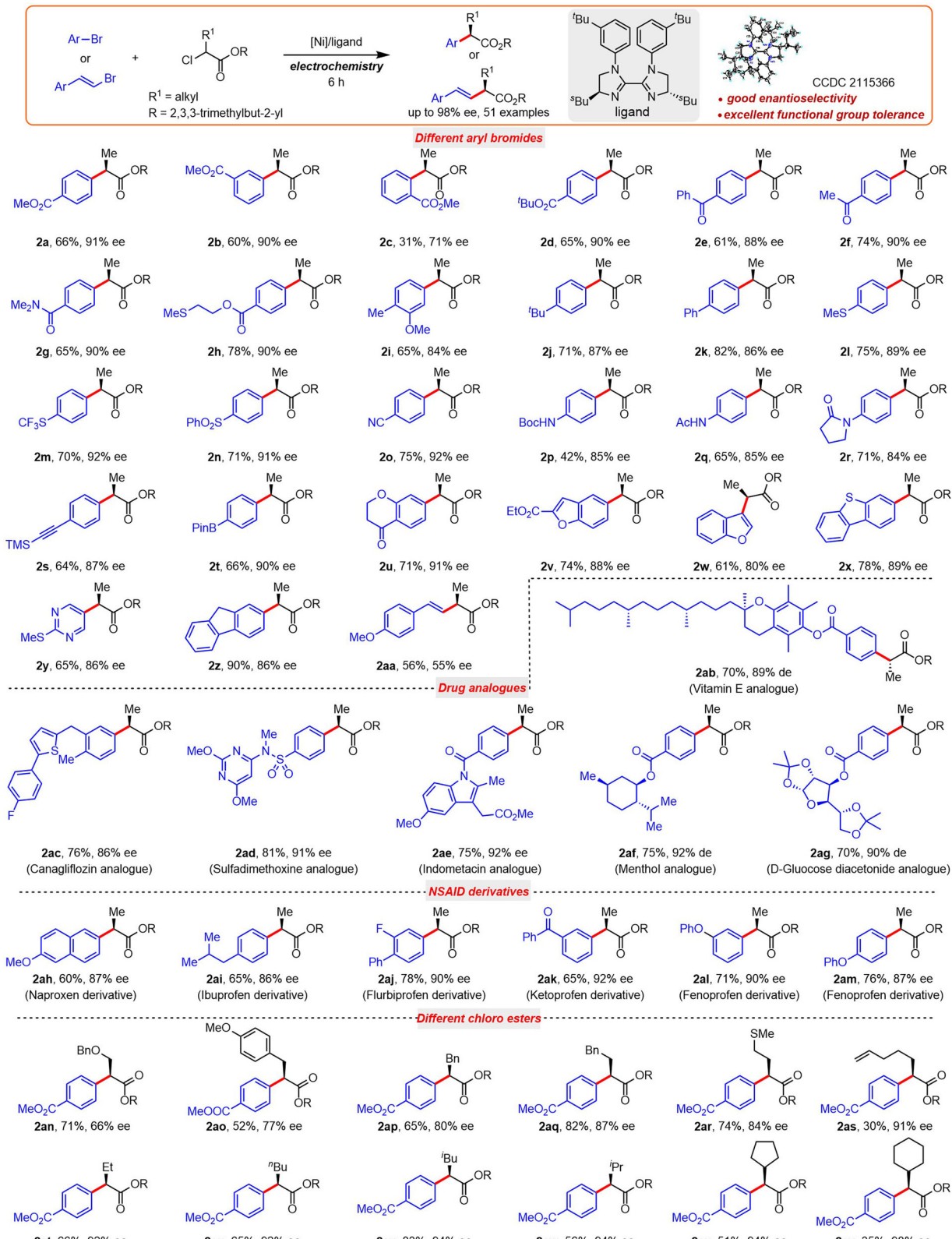

**Fig. 2 | Evaluation of substrate scope.** Yield of isolated products unless otherwise indicated. The reaction was carried out in an undivided cell with **1a** (0.2 mmol), (*rac*)-**1b** (2.0 equiv), NiBr₂•glyme (10.0 mol%), **L7** (15.0 mol%), DMAc (0.1 M), *n*-Bu₄NBF₄ (1.0 equiv), TTMSS (3.5 equiv), 2,6-lutidine (3.0 equiv), platinum anode

(1.0 cm × 1.0 cm) and Ni foam cathode (2.0 cm × 3.0 cm), constant potential (U_cell = 2.9 V, 6.0 h for 0.2 mmol scale), rt. Enantioselectivities were determined by chiral HPLC analysis. TTMSS stands for Tris(trimethylsilyl)silane.

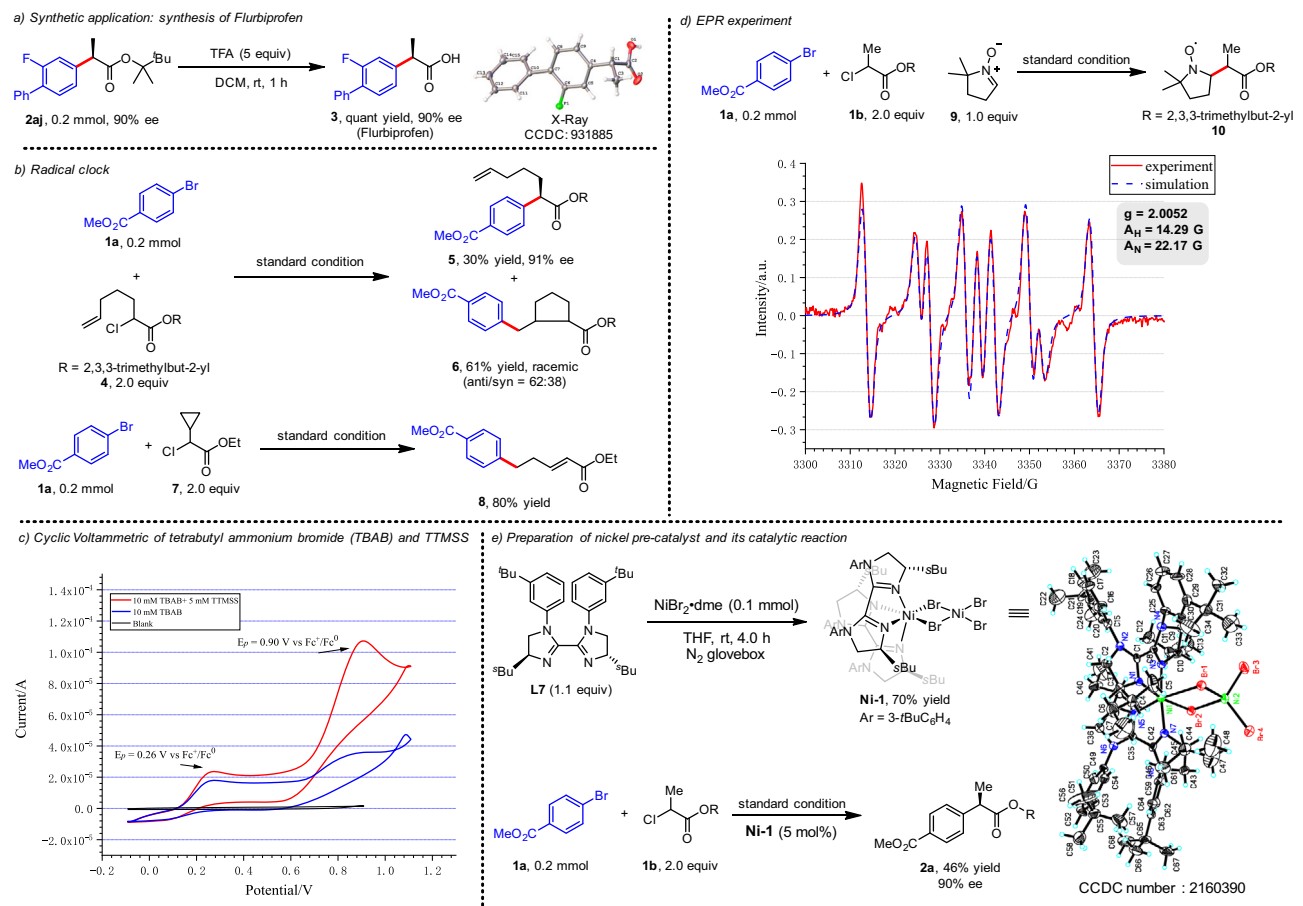

**Fig. 3 | Synthetic application and mechanistic studies. a** Synthetic application: synthesis of flurbiprofen. **b** Radical clock. **c** Cyclic voltammetric of tetra-butyl ammonium bromide (TBAB) and TTMSS. **d** EPR experiment. **e** Preparation of nickel precatalyst and its catalytic reaction.

these Ni-catalyzed asymmetric electrochemical arylations (as shown in Fig. 2). The position of substituents has a strong influence on both yield and enantioselectivity of the coupling products (**2a**–**2c**). Excellent functional group tolerance was observed, including ester (**2d**), ketone (**2e**, **2f**), amide (**2g**), methylthio (**2 h**, **2 l**), ether (**2i**), alkyl (**2j**), as well as aryl (**2k**) substituents. Aryl bromides with electron-withdrawing groups (**2m**–**2r**) exhibited good enantioselectivity (84–92%). It is noteworthy that the trimethylsilyl and boronate ester were also both well accommodated under electrochemical conditions, affording the products **2s** and **2t** in 64 and 66% yield with 87 and 90% ee, respectively. Heterocycles such as chromanone, benzofuran, benzothiophene, and pyrimidine proved to be good coupling partners, affording the corresponding products **2u**–**2y** in good yield and 86–91% ee. For other evaluated heteroaryl bromides, low conversion of 3-bromoquinoline, 3-bromopyridne, 6-bromobenzo[d]oxazole, and 2-bromo-1H-imidazole were observed (Supplementary Table 8). Excellent yield and good enantioselectivity were observed when 2-bromofluorene was employed (**2z**). An alkenyl bromide afforded moderate yield and diminished enantioselectivity of the corresponding product **2aa**. Unfortunately, aryl iodide, chloride, and alkenyl triflates were unsuccessful under present reaction conditions (see Supplementary Table 8 for details).

To illustrate the synthetic utility of our protocol, we applied it to the preparation of α-aryl ester derivatives of analogs of several known medicines, including those of canagliflozin, sulfadimethoxine, indomethacin, (–)-menthol, and D-glucose diacetonide, affording the corresponding products **2ab**–**2ag** in 70–81% yield and 86–92% enantioselectivity. Furthermore, six well-known NSAID derivatives (**2ah**–**2am**) were readily synthesized using our method. These results

demonstrate the breadth of opportunity available for modern applications in drug discovery and agrochemical synthesis.

The influence of the α-alkyl substituent of the α-chloro ester reactant was also probed. Ether-bearing esters afforded either lower enantioselectivity or yield (**2an** and **2ao**, respectively). A benzyl substituent afforded attenuated enantioselectivity (**2ap**), whereas a homobenzyl substituent fared much better in terms of both yield and enantioselectivity (**2aq**). Methanethio fared marginally better than benzyl in terms of both yield and enantioselectivity (**2ar**). Enantioselectivities ranged from 91–98% for non-arene-bearing hydrocarbon substituents, though yields varied widely (**2as**–**2ay**).

## Discussion

Flurbiprofen was readily prepared by quantitative hydrolysis of **2aj**, and was found to have (R) configuration by X-ray analysis (Fig. 3a). To gain insight into the mechanism of this electrochemical cross-coupling, we reacted **1a** with radical clocks **4** and **7**. Cyclized product **6** was obtained from **4** as a racemic mixture of diastereomers (61% yield, trans:cis = 62:38) (Fig. 3b). When cyclopropyl-containing **7** was employed, ring-opened product **8** was formed with 80% yield. These results suggest that this nickel-catalyzed cross-coupling process likely involves a radical pathway.

Next, we probed the electrochemical cross-coupling reaction mechanism via cyclic voltammetric analyses (Fig. 3c). A 0.1 M solution of TBAB in DMAc, exhibits oxidation peaks at 0.26 and 0.90 V versus $Fc^+/Fc^0$ (see Supplementary Fig. 2 for details). An increase in the second oxidation peak was observed after TTMSS (5 mM) was added. This result indicates that the bromide anion is converted to an electrophilic bromine radical following oxidation; bromine radicals are known to

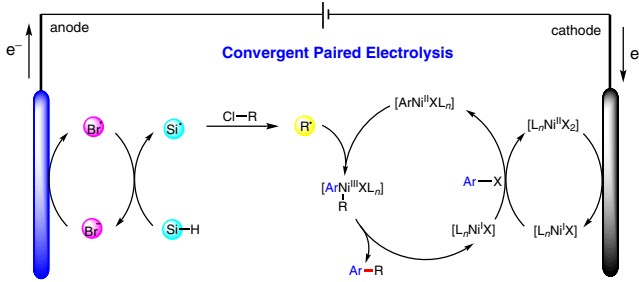

**Fig. 4 | Plausible mechanism.** The electrophilic alkyl radical from the α-carbonyl chloride is formed via XAT and HAT processes in the anode. Cathodic reduction of $Ni^{II}X_2L_n$ species leads to $Ni^IXL_n$ species, which undergoes the oxidative addition of aryl halides. The capture of the alkyl radical by $Ni^IXL_n$ species delivers an (alkyl) (aryl)$Ni^{III}XL_n$ species, then rapid reductive elimination yields product **2** and regenerates the $Ni^I$ species.

rapidly abstract a hydrogen atom from the silane. The silyl radical can then rapidly abstract a chlorine atom from the activated α-carbonyl chloride to generate an electrophilic alkyl radical species[66,67]. The alkyl radical was trapped by DMPO (5,5-dimethyl-1-pyrroline *N*-oxide) and the resulting more stable radical **10** was detected by EPR (electron paramagnetic resonance, Fig. 3d). Finally, nickel precatalyst (**Ni-1**) was synthesized by reacting NiBr₂•dme and chiral BiIM ligand **L7** at room temperature for 4 h, and its structure was verified by X-ray analysis (Fig. 3e). Interestingly, hexa-coordinated and tetra-coordinated nickel species constitute this binuclear nickel complex **Ni-1**. We performed the reaction with **Ni-1** as the catalyst, the desired product **2a** was obtained in 46% yield with 90% ee (Fig. 2e). Although **Ni-1** provides good reactivity and enantioselectivity in this transformation, the 1:1 coordination of nickel and BiIM as catalytic activity species cannot be ruled out. Based on the mechanistic experiments above, and the related studies by Diao[68,69,85,86], we propose the putative mechanism shown in Fig. 4. Initially, the anode oxidizes the bromide anion to a bromine radical, which can rapidly abstract a hydrogen atom from the silane. The resulting silyl radical can abstract the chlorine atom from the α-carbonyl chloride to generate an electrophilic alkyl radical species. The $Ni^{II}X_2L_n$ species is reduced to a $Ni^IXL_n$ species by cathodic reduction. Then, following oxidative addition in the presence of another $Ni^IXL_n$ species, an $ArNi^{II}XL_n$ intermediate and $Ni^{II}X_2L_n$ are generated[68,69]. Oxidative capture of an α-acyl radical could then deliver an (alkyl)(aryl)$Ni^{III}XL_n$ species, which then undergoes rapid reductive elimination to form the C($sp^2$)–C($sp^3$) bond of **2** and regenerate the $Ni^I$ species. However, a $Ni^0$/$Ni^{II}$/$Ni^{III}$/$Ni^I$ catalysis sequence (rather than the aforementioned $Ni^I$/$Ni^{II}$/$Ni^{III}$ sequence) cannot be ruled out at this stage.

In summary, we develop a paired electrolysis-enabled Ni-catalyzed enantioselective reductive cross-coupling of aryl bromides and α-chloro esters under mild reaction conditions, affording enantioenriched α-aryl esters with good yields. Additionally, various nonsteroidal anti-inflammatory drug (NSAID) derivatives can be efficiently synthesized, and numerous drug analogs can be diversified through these asymmetric electrochemical reductive couplings. In this electrochemical process, alkyl radicals are generated from alkyl halides through a sequential electron transfer process involving anodic oxidation, while at the same time, the nickel catalyst is reduced to a lower oxidation state via cathodic reduction. Preliminary mechanistic experiments strongly support the synergistic involvement of anodic and cathodic redox events.

## Methods
### General procedure for the electrolysis
In a nitrogen-filled glove box, an over-dried 10 mL hydrogenation tube charged with a stir bar, NiBr₂•glyme (10.0 mol%, 6.2 mg), **L** (15.0 mol%,

16.0 mg). TBABF₄ (62.0 mg, 0.2 mmol) and DMAc (1.0 mL) were added to the electrochemical cell and the mixture was stirred for over 20 min. Then aryl bromide (0.2 mmol), 2,3,3-trimethylbutan-2-yl-2-chloropropanoate (90.0 mg, 0.4 mmol), 2,6-lutidine (62.0 mg, 0.6 mmol), TTMSS (175.0 mg, 0.70 mmol), and DMAc (1.0 mL) were added to the electrochemical cell. The tube was installed with Ni foam (2.0 cm × 3.0 cm) as the cathode and Pt (1.0 cm × 1.0 cm) as the anode. The seal tube was sealed and removed from the glove box. The reaction mixture was electrolyzed under a constant current of 2.9 V until the complete consumption of the starting materials was judged by TLC (about 6 h). After the reaction, the aqueous layer was extracted with EtOAc (3 x equal volume) and the combined organics were washed with sat. brine (4 x equal volume), dried over MgSO₄, filtered and concentrated in vacuo. The crude product was purified by column chromatography to furnish the desired product.

More experimental procedures and a photographic guide for Ni-catalyzed enantioselective electrochemical reductive cross-couplings are provided in the Supplementary Information.

## Data availability
The X-ray crystallographic coordinates for structures reported in this article have been deposited at the Cambridge Crystallographic Data Center (CCDC), under deposition number CCDC 2115366 (**L7**), CCDC 931885 (**3**), and CCDC 2160390 (Ni-**1**). The data can be obtained free of charge from the Cambridge Crystallographic Data Center [http://www.ccdc.cam.ac.uk/data_request/cif]. The data supporting the findings of this study are available within the article and its Supplementary Information files. Any further relevant data were available from the authors on request.

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

## Acknowledgements

This work was financially supported by the National Key R&D Program of China (No. 2021YFA1500100), the NSF of China (Grants 21821002, 21772222, and 91956112), the S&TCSM of Shanghai (Grants 18JC1415600 and 20JC1417100), Bayer AG (Germany), and the fellowship of China Postdoctoral Science Foundation (2020M671274).

## Author contributions

D.L., Z.-R.L., and Z.-H.W. designed and performed the experiments. T.-S.M. directed the project. C.M., S.H., and H.S. revised the manuscript. D.L., Z.-H.W., and T.-S.M. wrote the manuscript with input from all authors. All authors analyzed the results and commented on the manuscript.

## Competing interests

The authors declare no competing interests.
