## [Peer Review File · Nature Communications]

REVIEWER COMMENTS

Reviewer #1 (Remarks to the Author):

Electrochemical asymmetric cross-coupling has received extensive attention as a green synthetic method. In this manuscript, Mei and co-workers have reported an efficient pathway for rapid construction of α -aryl esters with high ee from accessible aryl bromides and α -chloroesters by a paired electrolysis-enabled Ni-catalyzed enantioselective reductive cross-coupling strategy. Meanwhile, the authors investigated the reaction mechanism through radical trapping experiment, cyclic voltammetry analysis, and EPR experiments. Although the similar arylation was reported by the Mao and Reisman group and the substrate scope is somehow limited, the electrochemical reductive cross-coupling of aryl bromides with α -chloroesters provides a concise protocol for the preparation of chiral phenylacetic acid and derivatives. Therefore, it is recommended to publish the manuscript in Nature communication before the following issues to be addressed.

1. Are the aryl iodides, the lower reactivity of aryl chloride and alkenyl triflates compatible with the reaction conditions? The authors should try to demonstrate the practicality of the method as much as possible.
2. Could heteroaryl bromides be tolerated, e.g. quinoline, pyridine, pyrazine, imidazole, benzoxazole?
3. In addition, have the authors tried α -bromoesters under the standard conditions?
4. The alkylation example (2aa) is similar to arylation but novel. Could the authors demonstrate more alkylation cases?
5. In the condition screening, the R group of the α -chloroesters has a significant influence on the ee value, is that caused by steric hindrance? How about using α -chloroesters derived from complex molecules?
6. The authors proposed bromide ions as HAT reagents to react with silane to generate silicon radicals. Have they captured bromine radicals? In table S1, when pyridine was used as the base, the desired product 2a was not obtained, but with 2,6-lutidine, the product was afforded in surprisingly 91% yield. What is the oxidation potential of pyridine and pyrimidine? Is it possible that 2,6-lutidine be oxidized to nitrogen radical cations under standard conditions to react with silanes as HAT reagents?
7. How about using Ni(COD)₂ as the catalyst?
8. Please unify the format of the uppercase and lowercase of the reference titles. In Supporting Information, the numbering of tables should be corrected, such as Table S1.

Reviewer #2 (Remarks to the Author):

The application of convergent paired electrolysis in asymmetric electro organic synthesis is noteworthy as it represents a highly original and atom-economical, efficient approach to accessing valuable chiral products.

The results presented in the manuscript are highly significant as they provide a platform for other groups to build from in applying this technique in (presumably) a broad range of asymmetric transformations. The use of a nickel catalyst is also a highlight as a 1st row (abundant) metal avoids many of the toxicity / cost issues associated with the heavier 2nd and 3rd row congeners.

The mechanistic and spectroscopic studies are consistent with the proposed mechanism and likely intermediates the authors have advanced. The methods are well described and should be reproducible by other workers in the field.

I'd also like to say this was a highly enjoyable article to read, was very well written and clear. It isn't often that I come across a paper with no major issues that have to be addressed, so I'd like to say thank you to the authors for that! Two very minor points are the used of "reductive reduction" - perhaps change to "cathodic reduction"? and I did see a very new review come out by Zhang in Chemistry A European Journal on convergent paired electrolysis it might be helpful to cite.

Otherwise, I believe this will be a strong addition to the field of organo eletrosynthesis which will be widely read and cited.

Response letter

Reviewer: 1

Referee 1 was highly enthusiastic and suggested “publish after minor revisions”. The reviewer offers a few suggestions.

Question 1: Are the aryl iodides, the lower reactivity of aryl chloride and alkenyl triflates compatible with the reaction conditions? The authors should try to demonstrate the practicality of the method as much as possible.

Our response: Thanks for your suggestion. We have investigated other substrates including aryl iodide, chloride, and alkenyl triflates. However, these substrates could not provide good yields of desired cross-coupling products at present reaction conditions, although some of them could give moderate to good enantioselectivity. We have included these results in the revised Supporting Information (Table S8). Additionally, we have included the following statement in the manuscript at the end of the first paragraph on page 6: “Unfortunately, aryl iodide, chloride and alkenyl triflates were unsuccessful under present reaction conditions (see Table S8 in the Supporting Information for details).”

Standard condition: aryl bromide/iodine/chloride (0.2 mmol), (rac)-1b (2.0 equiv), NiBr₂·glyme (10.0 mol%), Ligand (15.0 mol%), DMAc (0.1 M), *n*-Bu₄NBF₄ (1.0 equiv), TTMSS (3.5 equiv), 2,6-lutidine (3.0 equiv), platinum electrodes (+) and Ni foam (-), constant voltage (U_{cell} = 2.9 V, 6.0 h for 0.2 mmol scale), rt, isolated yield.

Question 2: Could heteroaryl bromides be tolerated, e.g. quinoline, pyridine, pyrazine, imidazole, benzoxazole?

Our response: Thanks for your suggestion. In the manuscript, 5-bromo-2-(methylthio)pyrimidine (**2y**) was well tolerated in our reaction conditions, which could provide the corresponding cross-coupling products in 65% isolated yield with 86% enantioselectivity. We have investigated other heteroaryl bromides including 3-bromoquinoline, 6-bromobenzo[*d*]oxazole, and 2-bromo-1*H*-imidazole. However, these substrates are not reactive and only lead to trace products. However, 3-bromopyridine gives the desired product in 30% yield and 80% enantiomeric excess. We have included these results in the revised Supporting Information (Table S8). Additionally, we have included the following statement in the manuscript at the middle of the first paragraph on page 6 : “For other evaluated heteroaryl bromides, low conversion of 3-bromoquinoline, 3-bromopyridine, 6-bromobenzo[*d*]oxazole, and 2-bromo-1*H*-imidazole were observed (Table S8 in the Supporting Information).”

Standard condition: aryl bromide (0.2 mmol), (*rac*)-1b (2.0 equiv), NiBr₂·glyme (10.0 mol%), Ligand (15.0 mol%), DMAc (0.1 M), *n*-Bu₄NBF₄ (1.0 equiv), TTMSS (3.5 equiv), 2,6-lutidine (3.0 equiv), platinum electrodes (+) and Ni foam (-), constant voltage (U_{cell} = 2.9 V, 6.0 h for 0.2 mmol scale), rt, isolated yield.

Question 3: In addition, have the authors tried α -bromoesters under the standard conditions?

Our response: Thanks for your suggestion. We have included one example with α -bromoester. Employment of α -bromo ester and methyl 4-bromobenzoate (**1a**), **2a** was obtained in 16% yield and 91% enantiomeric excess. The perfect match of the rate of

the anodic and cathodic processes is critical for the high yield of the product. The low yield of the product is mainly due to the silyl radical abstract the Br atom is fast than Cl atom, which causes the alkyl radical formation is much faster in the case of α -bromo esters than α -chloro esters. These results were added to Table S8 in the revised Supporting Information. Additionally, The statement “Replacement of α -chloro esters with α -bromoesters resulted in a low yield of the desired product, but good enantioselectivity (91%) was obtained (Table S8 in the Supporting Information).” was added at the end of Optimization Studies (page 5).

Question 4: The alkylation example (**2aa**) is similar to arylation but novel. Could the authors demonstrate more alkylation cases?

Our response: Thanks for your great suggestion. We carried out the reactions with (*E*)-(2-bromovinyl)benzene and 2-bromo-1H-indene as substrates, moderate yields were obtained along with 52% and 72% enantioselectivity, respectively. These results were included in the Table S8 in the revised Supporting Information.

Question 5: In the condition screening, the R group of the α -chloroesters has a significant influence on the ee value, is that caused by steric hindrance? How about

using α -chloroesters derived from complex molecules?

Our response: Thanks for your comments. In this manuscript, we were pleased to find that the steric hindrance at the α -position of chloro has a significant influence on the ee value (**2au**, **2av**, **2ax**, **2ay** in the manuscript). When using α -chloroesters derived from complex molecules, such as the menthol analog, only moderate dr was obtained. These results were added to Table S8 in the revised Supporting Information.

Question 6: The author's proposed bromide ions as HAT reagents to react with silane to generate silicon radicals. Have they captured bromine radicals? In table S1, when pyridine was used as the base, the desired product **2a** was not obtained, but with 2,6-lutidine, the product was afforded in surprisingly 91% yield. What is the oxidation potential of pyridine and pyrimidine? Is it possible that 2,6-lutidine be oxidized to nitrogen radical cations under standard conditions to react with silanes as HAT reagents?

Our response: Thanks for this suggestion. We have carried out experiments with the addition of alkene or alkyne under standard conditions. Unfortunately, there is no bromine radical addition product was observed.

Additionally, we carried out the CV study with pyridine and 2,6-lutidine, the oxidation potential of pyridine and 2,6-lutidine is 1.85 V and 1.9 V vs Fc^+/Fc^0 , respectively. Both of this oxidative potential is much higher than Br anion (maximum oxidative potential is 0.9 V). Therefore, Br anion would prefer oxidized at anode rather than pyridine and pyrimidine. The CV chart was added to Figure S3 in the revised Supporting Information.

Question 7: How about using Ni(COD)₂ as the catalyst?

Our response: We have carried out experiments with Ni(COD)₂ as the catalyst under the standard conditions, it affording **2a** in 46% yield and 91%. This result was added to entry 11, Table S7 in the Supporting Information.

Question 8: Please unify the format of the uppercase and lower case of the reference titles. In Supporting Information, the numbering of tables should be corrected, such as Table S1.

Our response: Thanks for this suggestion. We have unified the format of the titles in uppercase as a highlighted in yellow in the revised manuscript. We have changed “Table S2” to “Table S1” in the revised Supporting Information.

Reviewer: 2

Referee 2 was highly enthusiastic and suggested “publish after minor revisions”. The reviewer offers a few comments.

Question 1: Two very minor points are the used of "reductive reduction" - perhaps change to "cathodic reduction"? and I did see a very new review come out by Zhang in Chemistry A European Journal on convergent paired electrolysis it might be helpful to cite."

Our response: Thanks for your suggestion, we have changed "reductive reduction" to "cathodic reduction" in the revised manuscript as highlighted in yellow.

In addition, we have cited the mentioned paper in References 32 in the revised manuscript. " Zhang, S. & Findlater, M. Progress in Convergent Paired Electrolysis. *Chem. Eur. J.* **28**, e202201152 (2022). "

REVIEWERS' COMMENTS

Reviewer #1 (Remarks to the Author):

Recommendation: Accept

Comments:

As the comments for the original manuscript was fully addressed, the manuscript is recommended for publication.